environmental science, plant science, ecology

plant-derived foods, insect community, multi-trophic interactions, biological control

**Author for correspondence:**
Pablo Urbaneja-Bernat
e-mail: pablo.urbaneja@rutgers.edu

# Plant guttation provides nutrient-rich food for insects

Pablo Urbaneja-Bernat[1], Alejandro Tena[2], Joel González-Cabrera[3] and Cesar Rodriguez-Saona[1]

[1]Department of Entomology, Rutgers University, P.E. Marucci Center, 125A Lake Oswego Road, Chatsworth, NJ, USA
[2]Instituto Valenciano de Investigaciones Agrarias (IVIA), Centro de Protección Vegetal y Biotecnología, Unidad Mixta Gestión Biotecnológica de Plagas UV-IVIA, Moncada, Valencia, Spain
[3]Department of Genetics, ERI-BIOTECMED, Unidad Mixta Gestión Biotecnológica de Plagas UV-IVIA, Universitat de València, Burjassot, Valencia, Spain

 PU-B, 0000-0002-6995-5468; AT, 0000-0002-5001-4334; JG-C, 0000-0002-8338-370X; CR-S, 0000-0001-5888-1769

Plant guttation is a fluid from xylem and phloem sap secreted at the margins of leaves from many plant species. All previous studies have considered guttation as a water source for insects. Here, we hypothesized that plant guttation serves as a reliable and nutrient-rich food source for insects with effects on their communities. Using highbush blueberries as a study system, we demonstrate that guttation droplets contain carbohydrates and proteins. Insects from three feeding lifestyles, a herbivore, a parasitic wasp and a predator, increased their longevity and fecundity when fed on these guttation droplets compared to those fed on control water. Our results also show that guttation droplets, unlike nectar, are present on leaves during the entire growing season and are visited by numerous insects of different orders. In exclusion-field experiments, the presence of guttation modified the insect community by increasing the number of predators and parasitic wasps that visited the plants. Overall, our results demonstrate that plant guttation is highly reliable, compared to other plant-derived food sources such as nectar, and that it increases the communities and fitness of insects. Therefore, guttation represents an important plant trait with profound implications on multi-trophic insect–plant interactions.

## 1. Introduction

Plants possess multiple characteristics that affect their interactions with mutualistic and antagonistic organisms [1,2], such as the plants' interactions with herbivores and the natural enemies of herbivores [3–7]. In the last two decades, ecologists have reinforced the importance of plant-derived products, such as pollen, floral nectar, extrafloral nectar and honeydew, as supplementary diets for the natural enemies of herbivores [8–10]. These plant-derived products constitute a rich source of carbohydrates and proteins that are essential for natural enemies that can control herbivorous pests [10–12]. However, many of these sources are ephemeral (e.g. pollen or floral nectar) or their quality as a carbohydrate source is variable (e.g. honeydew excreted by hemipterans) [10,11,13]. Therefore, these plant-derived products are not always dependable food sources for natural enemies, and their absence or low quality can affect the regulation of herbivores, with consequent negative effects on plants.

Plant guttation is an exudation fluid (figure 1*a*) that is secreted at the margins and tips of leaves through pores, known as hydathodes, in the form of droplets [14–16]. These guttation droplets are composed of xylem and phloem sap [15–17], and occur in a wide range of plant species [18–22] including rice [23], wheat [24], barley [24], rye [24], oats [25], sorghum [26], maize [27], tobacco [28], tomato [29], strawberry [30], cucumber [30] and blueberry [31]. The presence of guttation on leaves is controlled by root pressure that it is affected by abiotic

**Figure 1.** Guttation in highbush blueberries, *V. corymbosum*. (*a*) Guttation droplets from a blueberry leaf. (*b*) A predatory cecidomyiid fly flying from a blueberry leaf after feeding on a guttation drop. (*c*) A predatory dolichopodid fly feeding on a guttation drop. (*d*) A crab spider (Thomisidae) preying on a dolichopodid fly near the guttation drops. (*e*) Ants (Formicidae) feeding on guttation drops. (*f*) A lacewing (Chrysopidae) adult feeding on a guttation drop. (*g*) A drosophilid fly feeding on a guttation drop. (Online version in colour.)

(i.e. ambient and soil temperatures, relative humidity (RH) and solar radiation) and biotic (i.e. vegetative and reproductive growth) factors [15–17,24,32,33]. Previous studies have considered guttation as a water source (referred to as 'guttation water') for insects and have, thus, ignored its potential beneficial effects as a source of nutrients. Furthermore, some of these studies focused on the potential negative effects of guttation on pollinators and natural enemies since it may be a potential route for pesticide exposure [34–38]. However, guttation is more than water. As plant exudate, guttation droplets can contain carbohydrates and proteins from the xylem and phloem [24,39]. Despite this evidence and to the best of our knowledge, no studies have evaluated guttation as a food source for insects and, more importantly, evaluated the effects of guttation drops on insect community composition. This is especially important because a large number of previous studies have dealt with the scarcity of food sources for natural enemies in ecosystems [10–12], and all these studies have overlooked the presence of guttation as a food source.

Here, we addressed these knowledge gaps by documenting for the first time the significance of plant guttation as a potential nutrient-rich food source for insects in ecosystems. In detail, we first tested (i) whether leaf guttation drops are a nutritious food source for insects that feed on it. To test (i), we measured the longevity and fecundity of insects from three feeding lifestyles, a herbivore, a parasitic wasp and a predator, and analysed the nutritional (sugar and protein) content of leaf guttation. Then, we tested (ii) whether leaf guttation drops are, unlike nectar, a reliable food source for insects in the field. To test (ii), we first measured the abundance of guttation drops not only throughout the season but also during the day because guttation depends on weather conditions that vary along the day. Then, we counted and identified insects that visited leaves with guttation drops. Finally, we tested (iii) whether guttation affects the insect community composition that visits plants. To test (iii), we identified and counted insects that visited plants with and without guttation drops in a manipulative study in the field.

## 2. Material and methods

### (a) Study system
We conducted our studies in highbush blueberry (*Vaccinium corymbosum* L.). Highbush blueberry is a crop native to North America with expanding production and consumption worldwide [40]. For our experimental assays under controlled conditions, we used adults of three insect taxa with different lifestyles that are common in blueberry fields, namely, a herbivore (the spotted-wing drosophila, *Drosophila suzukii* (Matsumura) (Diptera: Drosophilidae)), a parasitic wasp (*Aphidius ervi* Haliday (Hymenoptera: Braconidae)) and a generalist predator (the green lacewing, *Chrysoperla rufilabris* (Burmeister) (Neuroptera: Chrysopidae)).

### (b) Plants and insects
Two-year-old potted highbush blueberries (*V. corymbosum*, cv 'Bluecrop'), free of pesticides, were used for laboratory and field experiments. Plants were maintained in a glasshouse for five months (April–August) at $20 \pm 2$°C, $70 \pm 10\%$ RH and 15 : 9 light : dark (L : D); were fertilized twice (on 19 April and 10 May 2019) with granulated fertilizer N : P : K (10 : 10 : 10) and were irrigated three times per week.

*Drosophila suzukii* adults were obtained from a laboratory colony initiated in 2013 from wild specimens captured in Atlantic County (NJ, USA) and maintained at the Rutgers P.E. Marucci Center (Chatsworth, NJ, USA). The colony was maintained on standard *Drosophila* artificial diet [41,42] in 50 ml polystyrene vials (Fisher Scientific, Nazareth, PA, USA) with approximately 15 ml of diet and closed with BuzzPlugs (Fisher Scientific). *Aphidius ervi* (APHIDIUSforce E®) and *C. rufilabris* (CHRYSOforce R®) were obtained from a commercial supplier (Beneficial Insectary Inc., Redding, CA, USA). Bottles containing approximately 250 aphid mummies parasitized by *A. ervi* and frames of *C. rufilabris* containing approximately 500 pupae were separately placed inside individual 25 × 25 cm methacrylate rearing cages until emergence. Emerged adults were provided with a piece of wet cotton wool as a water source. All insects were maintained in an incubator (Percival Scientific, Perry, IA, USA) at $25 \pm 1$°C, $60 \pm 5\%$ RH and 16 : 8 h L : D.

### (c) Guttation collection
We collected guttation droplets manually from approximately 40 glasshouse-grown blueberry plants, from 15 May to 15 June 2019.

Groups of approximately five drops (4.8 ± 0.07) were removed from leaves by gently touching them with a $1 \times 1$ cm$^2$ piece of Parafilm® (Sigma-Aldrich®, St Louis, MO, USA) to allow the drops to adhere to the piece. The Parafilm® pieces were then stored at $-70°$C in Petri dishes labelled by date until used in experiments.

In addition, we estimated the amount of guttation provided to *D. suzukii*, *A. ervi* and *C. rufilabris* in the longevity and egg load experiments (see below) to ensure that all insects received similar amounts. The volume of each droplet was estimated as $(4/3 \times \pi \times r^3) \times 1/2$, where $r$ is the radius of the droplet. Based on this formula, the total daily amount of guttation offered to insects per each Parafilm® piece (1 cm$^2$), equivalent to approximately 5 guttation drops, was estimated at 0.28 ± 0.2 µl ($N$ = approx. 1000 drops or 200 Parafilm® pieces).

## (d) Guttation as a food source for insects

To test whether leaf guttation drops are a nutritious food source for insects that feed on it, we measured the adult longevity and fecundity of three insect taxa from different feeding lifestyles and analysed the nutritional (sugar and protein) content of leaf guttation. The effect of guttation on insect longevity was tested on three important insect taxa visiting guttation in the field (accounting for 40% of all visits; see Results): herbivores (*D. suzukii*, Drosophilidae), predators (*C. rufilabris*, Chrysopidae) and parasitic wasps (*A. ervi*, Braconidae). To evaluate the effect of guttation on the longevity of *D. suzukii*, *A. ervi* and *C. rufilabris* adults, we tested five diets: (i) water-only, (ii) sugar-only (1 M sucrose), (iii) protein-only (1 M yeast extract), (iv) sugar plus protein (1 : 1), and (v) guttation drops (which contains both sugars and proteins; see Results section). Sugar and protein diets were used as control diets to confirm their positive effects on the longevity and fecundity of the three insects. The sugar and protein concentrations used in diets 2, 3 and 4 were selected based on previous studies [43,44]. For each species, newly emerged (within 12 h of emergence) females and males were placed individually in 25 ml glass vials with the corresponding diet offered ad libitum. Food sources were provided with Parafilm® pieces and renewed daily. In the treatment of guttation drops (diet 5), Parafilm® pieces with five drops were defrosted at room temperature and provided to the insects. Water was provided in cotton balls that were renewed daily. Survival was checked daily until the insect died. There were 20 replicates for each diet and sex combination, and the experiment was conducted in a climatic chamber at 24.5 ± 1°C, 75 ± 5% RH and 16 : 8 h L : D.

To evaluate the effect of guttation on *D. suzukii*, *A. ervi* and *C. rufilabris* fecundity, approximately 75 individuals (2 : 1 female : male) of each insect species were placed inside a transparent, cylindrical polypropylene plastic cup (946 ml; diameter, 114 mm; height, 127 mm; Paper Mart, CA, USA) and were provided one of the five diets (ad libitum) described above for 24 h to ensure mating. After 24 h, 15 females per insect species and diet were killed and maintained at $-20°$C. Another 30 females per insect species and diet were individually placed in 25 ml glass vials, with the corresponding diet offered ad libitum. Food and water were provided as in the longevity assay (see above). After 3 and 7 days, females ($N$ = 15 per species/diet/day) were killed at $-20°$C. Therefore, we had females that were 1, 3 and 7 days old. To count the eggs, all frozen females were dissected by placing them on a microscope slide with a drop of water under a coverslip. To expose the ovaries with mature eggs out of the abdomen, we lightly applied pressure on the thorax with a pair of pins. The ovaries were photographed using a digital colour camera (Leica DFC500; Leica Microsystems Inc., Buffalo Grove, IL, USA) attached to a stereomicroscope (Leica MZ16), and the number of mature eggs was recorded.

## (e) Guttation chemical composition

The concentrations of total sugars and proteins were estimated in the abovementioned guttation samples that were collected and measured. The overall concentration of sugars in the guttation droplet was determined using a quantitative anthrone assay [45] with modifications. For this assay, we followed the same collection method of guttation drops explained above (Guttation collection). To analyse the guttation content, we processed 10 samples (each containing 126.4 ± 11.8 (mean ± s.e.) drops; each drop had a diameter of 0.46 ± 0.02 mm and a volume of 6.89 ± 0.31 µl) by adding 150 µl of phosphate-buffered saline (8 mM $Na_2HPO_4$, 2 mM $KH_2PO_4$ and 150 mM NaCl (pH 7.4)) to each sample to dilute the guttation, but further dilutions of the samples were made if needed depending on the sugar concentrations. Ten microlitres of each diluted sample were placed in a 96-well plate and mixed with 90 µl of anthrone reagent (1.5 mg ml$^{-1}$ in 95% sulfuric acid). The plate was stirred for 10 s and incubated for 2 h at 65°C (hot anthrone test). Then, the absorbance at 620 nm was measured in a 96-well plate reader (Tecan Infinite M200Pro; Tecan Austria GmbH, Grödig, Austria) and the concentration of total sugars estimated using known concentrations of glucose as standards.

The quantification of total proteins in the same 10 samples of guttation was performed by fluorimetry using the Qubit 3.0 fluorometer (Thermo Fisher, Hercules, CA, USA) at the proteomics section of the Central Service for Experimental Research, Universitat de València (Spain). One microlitre of each sample was examined using the Qubit Protein Assay Kit (Thermo Fisher, Hercules, CA, USA) according to the manufacturer's recommendations.

## (f) Reliability of guttation as a food source in the field

To test whether leaf guttation drops are a reliable food source for insects in the field, we measured the abundance of guttation drops throughout the season in two fields and counted and identified the insects that visited the leaves with guttation drops. Two highbush blueberry (*V. corymbosum* cv. 'Bluecrop') fields were selected to study the diurnal and seasonal occurrence of guttation droplets under field conditions. Fields were located at the Rutgers P.E. Marucci Center (Chatsworth, NJ, USA), were fertilized twice at the beginning of the season (on 2 May and 21 May 2019) with a granulated fertilizer N : P : K (10 : 10 : 10) and received water only by precipitation. No insecticides were applied to these fields. Each field consisted of 12 rows of 30 bushes per row (i.e. 360 bushes), with planting distances of 3 m between rows and 1.5 m between bushes within rows. At each field, we randomly selected 10 bushes from different rows (total $N$ = 20 bushes). In each bush, we labelled four stems with approximately 20 leaves each; two stems from the bottom half of the bush and two from the upper half of the bush at opposite orientations (north and south). To assess the presence of guttation, we selected five leaves randomly from each labelled stem and counted the total number of leaves with guttation and the number of guttation drops per leaf. In addition, we recorded the number and identity (to family) of the insects that were visiting guttation drops at the time of sampling. For each field, each selected bush was considered a replicate, and data on guttation and insect visitation were taken weekly from 2 May (early bloom) to 17 July (fruit maturation) of 2019 for a total of 12 sampling dates. On each sampling day, we sampled in the morning (08.00 h), at midday (13.00 h) and in the evening (18.00 h).

To determine the effects of abiotic factors on the occurrence of guttation droplets, data for daily maximum, minimum, and average ambient and soil temperatures; RH; and solar radiation were taken at 08.00, 13.00 and 18.00 h from 2 May until 17 July 2019 from a weather station located at the study site (Chatsworth, NJ, USA) (Office of the New Jersey State Climatologist; Rutgers

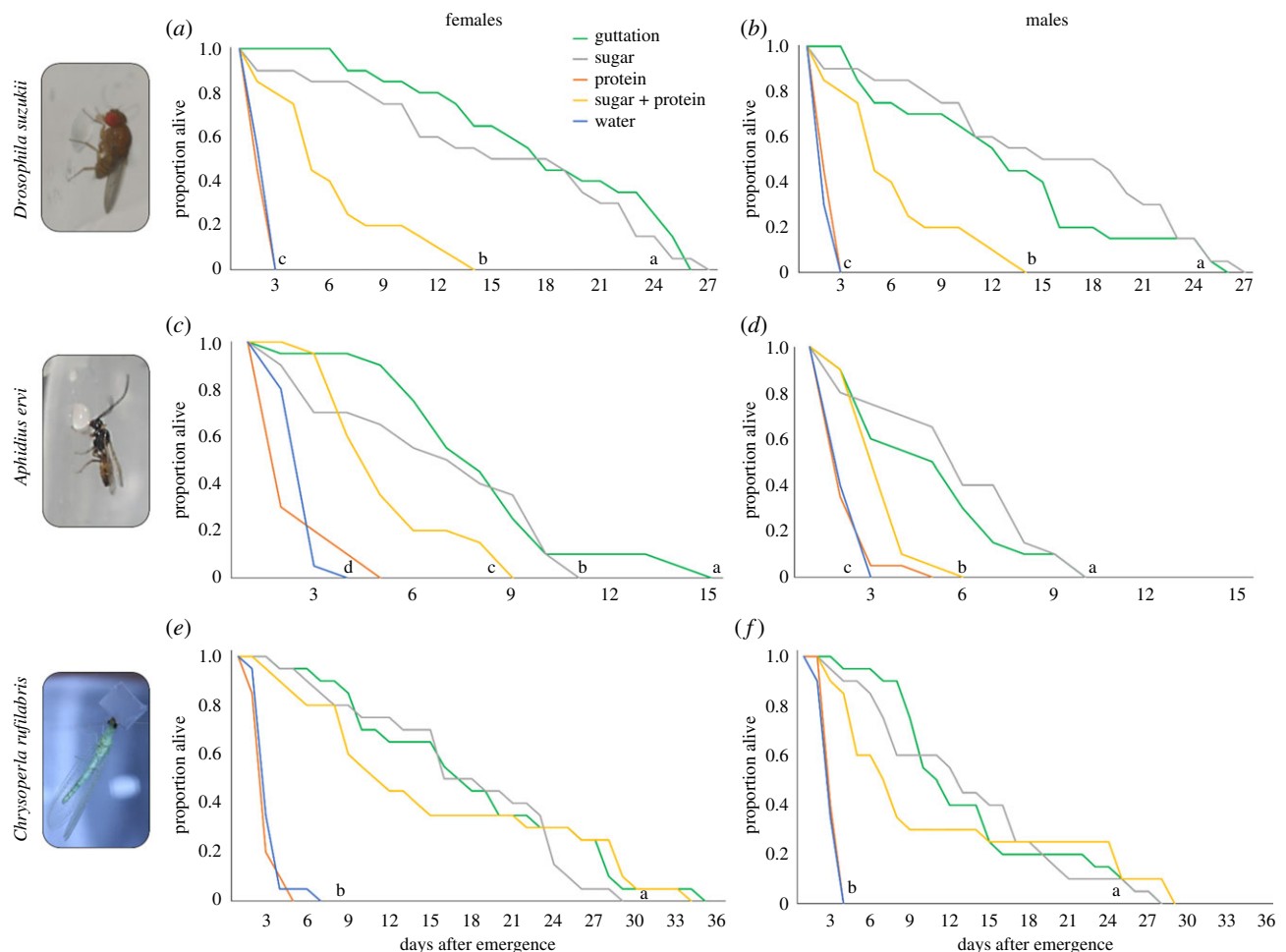

**Figure 2.** Survival curves for the spotted-wing drosophila *D. suzukii* (*a,b*), the braconid *A. ervi* (*c,d*) and the green lacewing *C. rufilabris* (*e,f*) females and males fed on five different diets: guttation, sugar-only, sugar plus protein, protein-only and water. Different letters indicate significant differences among diet treatments according to log-rank test of equality. (Online version in colour.)

University, New Brunswick, NJ, USA). The weather data were downloaded from the New Jersey Weather and Climate Network website (https://www.njweather.org/data) (electronic supplementary material, figure S1).

### (g) Guttation effects on insect community composition

To test whether guttation affects the insect community composition, we identified and counted insects that visited plants with and without guttation drops in a manipulative study. This study was conducted in one of the two blueberry fields mentioned above to identify the insect fauna attracted to the guttation drops by using flight interception traps. We used 10 glasshouse-grown potted blueberry plants (see above); leaves from five of these plants had guttation drops, whereas leaves from the other five plants had no guttation (all guttation droplets were removed manually before the experiment in the 'no guttation' plants). These potted plants were replaced with new ones daily to ensure the presence and absence of guttation in both treatments. The plants were placed in five different rows within the field (distance between rows = 6 m); two plants, one of each treatment (i.e. with or without guttation), were placed in each row. These two plants were placed close to the centre of the row and at least 10 m apart from each other. In each plant, we placed a transparent plastic card (10 cm × 30 cm) coated on both sides with Tangle-Trap (Tanglefoot Company, Bozeman, MT, USA). Each sticky trap was hung approximately 3 cm away from the apical shoots to capture the flying insect fauna attracted to, and leaving from, plants. For all plants, the number of leaves with guttation and the number of guttation drops per leaf were recorded. Traps were also replaced daily with new ones, and the experiment was run for 8

consecutive days. To avoid competition and increase insect attraction, the experiment was conducted at the time when leaves had hardened and guttation drops were almost completely absent in the field (i.e. post-harvest; 16 August–24 August 2019). Each potted plant was considered a replicate, and the insects on traps were identified to family and counted under a stereomicroscope (Leica MZ 9.5; Leica Microsystems, Inc., Wetzlar, Germany).

### (h) Statistical analysis

The Kaplan–Meier survival analysis, followed by a log-rank test of equality, was used to test for differences among survival curves for the five different diets for each insect species; each sex was analysed separately. We used generalized linear models assuming a Poisson distribution and log link function to determine differences in the number of mature eggs among diets for each insect species at different days after emergence.

We analysed field data with generalized linear mixed models. We also used a binomial distribution with logit link function to analyse the percentage of blueberry leaves with guttation presence and used a Poisson distribution with log link function for the number of guttation drops per leaf. Both models included 'date' (sampling date), 'time' (08.00, 13.00 and 18.00 h), and their interaction as fixed factors; 'field' (two blueberry fields) as a blocking factor and 'bush' (replicate) as a random factor. Correlation analyses (Pearson's coefficient) were conducted between each abiotic factor (i.e. RH, solar radiation, and ambient and soil temperatures) and the mean percentage of leaves with guttation drops present at the time of sampling (08.00, 13.00 and 18.00 h). Principal component analysis (PCA) was performed to visualize, through score and loading plots, differences in the insect community

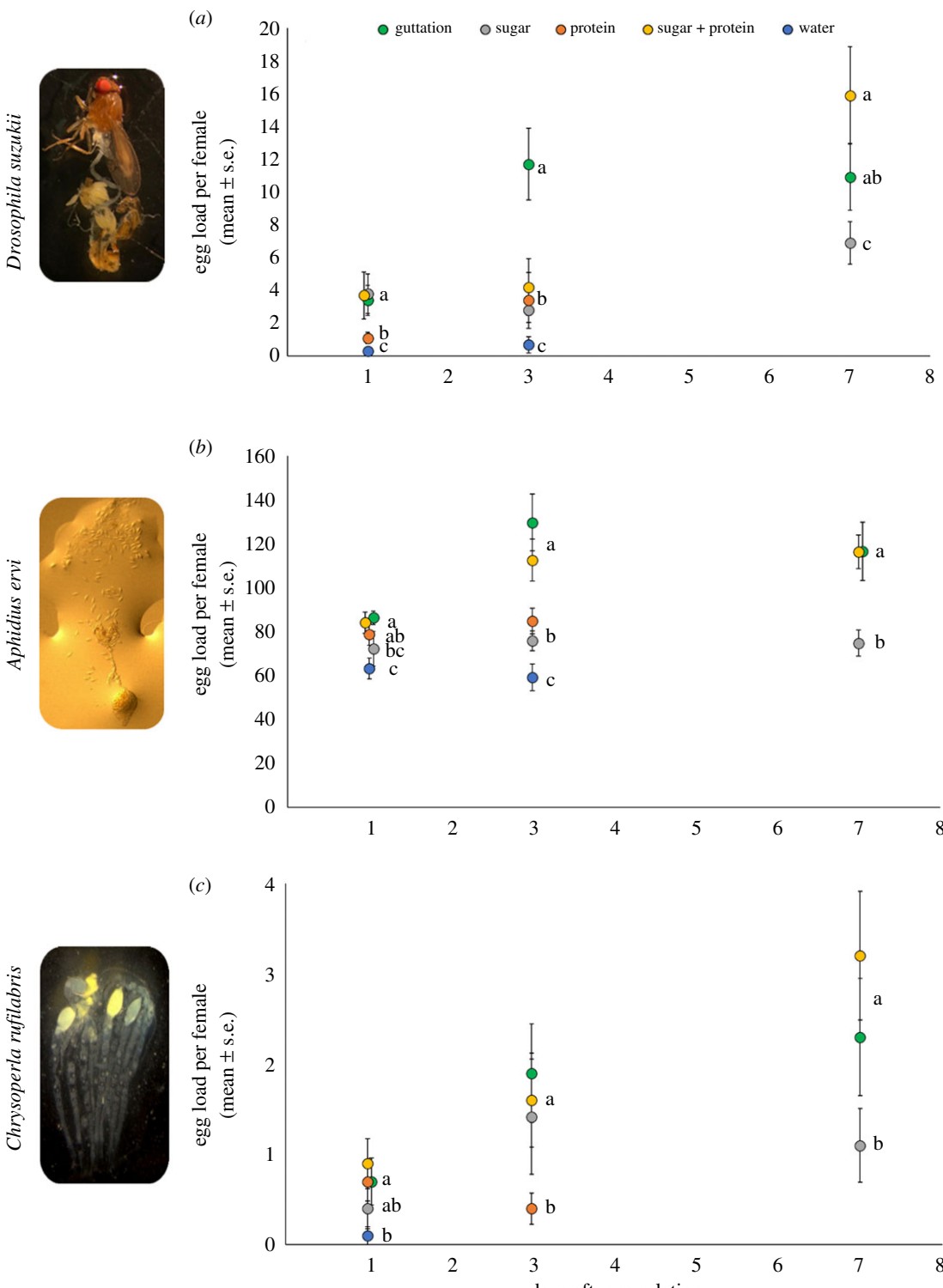

**Figure 3.** Number (±s.e.) of mature eggs produced by *D. suzukii* (*a*), *A. ervi* (*b*) and *C. rufilabris* (*c*) fed on five different diets: guttation, sugar-only, sugar plus protein, protein-only and water. Counts were taken 1, 3 and 7 days after adult emergence. Different letters indicate significant differences among diet treatments according to the Bonferroni pairwise tests. (Online version in colour.)

captured on sticky traps near potted blueberry plants with guttation versus those without guttation. The score plot was used to reveal the clustering of the insect communities between the two treatments (guttation versus no guttation), whereas the loading plot was used to display the contribution of each insect taxa to this clustering. Differences in the number of individuals of the main insect families captured between control (no guttation) and guttation traps were analysed using *t*-tests.

All statistical analyses were performed using IBM SPSS Statistics 23.0, except for PCA which was done using Minitab v. 16 (Minitab 2010).

## 3. Results

### (a) Guttation as a food source for insects: effects on insect longevity and egg load and composition

The longevity of adult females and males of *D. suzukii*, *A. ervi* and *C. rufilabris* was higher when they fed on guttation as a food source than on the sugar plus protein, protein-only or water-only diets but was similar to those that fed on a sugar-only diet (figure 2*a–f*; electronic supplementary

*Proc. R. Soc. B* **287**: 20201080

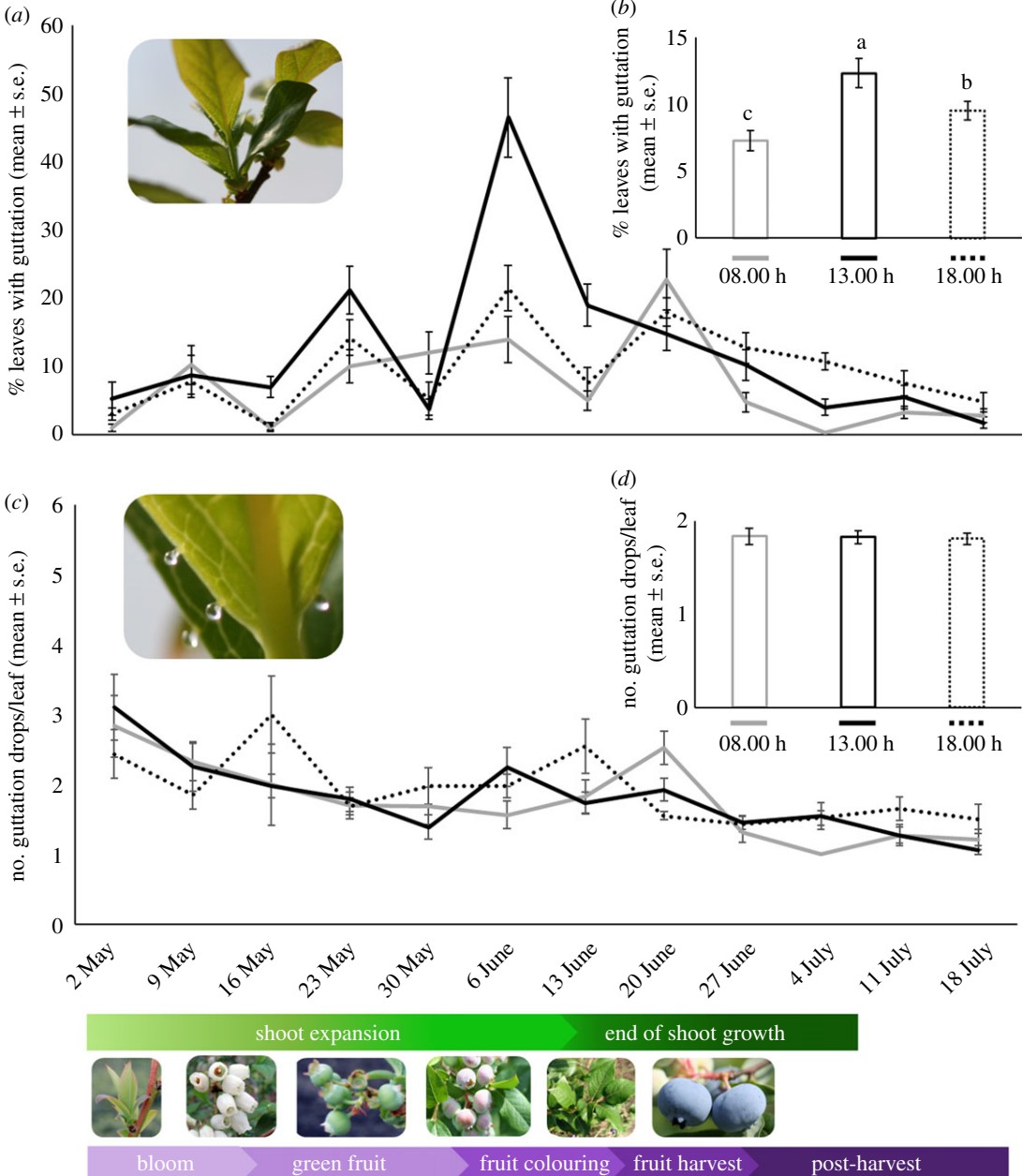

**Figure 4.** Occurrence of guttation drops during the highbush blueberry (*V. corymbosum*) growing season and at 08.00, 13.00 and 18.00 h under field conditions. (*a*) Percentage (mean ± s.e.) of leaves with guttation drops. Photo inset shows blueberry leaves. (*c*) Number of guttation drops per leaf (mean ± s.e.). Photo inset shows blueberry leaf with guttation drops. (*b,d*) Season totals (mean ± s.e.). Different letters above columns indicate significant differences at $\alpha = 0.05$ according to the Bonferroni pairwise tests. (Online version in colour.)

material, table S1). No individuals of any species survived more than 3 days on the protein and water-only diets.

In general, mated females of the three insect species had a higher egg load when they were given a guttation or a sugar plus protein diet than a sugar-only, a protein-only or a water-only diet 1, 3 and 7 days after emergence (figure 3*a*–*c*; electronic supplementary material, table S2).

The total amount of sugars and proteins per guttation droplet was estimated at $1.5 ± 0.2\ \mathrm{g\ ml}^{-1}$ (mean ± s.e.) and $4.3 ± 0.4\ \mathrm{mg\ ml}^{-1}$, respectively.

## (b) Reliability of guttation as a food source for insects in the field

In highbush blueberry fields, leaf guttation was present throughout the day and during the entire growing season (figure 4; electronic supplementary material, table S3). The highest percentage of leaves with guttation was observed during the phenological stage of shoot growth and expansion and early fruit development (green fruit and beginning of fruit colouring), when at noon more than 40% of the leaves had guttation drops (figure 4*a*). Drops per leaf decreased from the ninth week until the end of the study, which corresponds to the end of shoot growth and the post-harvest period (figure 4*c*). Among abiotic factors, we found that guttation in highbush blueberries was positively correlated with ambient and soil temperatures but not with RH or solar radiation (electronic supplementary material, figure S2 and table S4).

Throughout the blueberry growing season, a broad variety of arthropod fauna visited guttation drops (electronic supplementary material, table S5). The five most common insect taxa, accounting for approximately three-fourths of all observed insects visiting the guttation droplets in highbush blueberry fields, were ants (Formicidae) (19%), vinegar flies (Drosophilidae) (17%), lacewings (Chrysopidae) (15%), crab spiders (Thomisidae) (14%) and parasitic wasps

(8%) (figure 1; electronic supplementary material, table S5). We observed several insect taxa from different feeding lifestyles visiting the guttation drops. For example, predators in the families Cecidomyiidae (gall midges), Dolichopodidae (long-legged flies), Syrphidae (hoverflies), Formicidae (ants), Chrysopidae (lacewings) and Thomisidae (crab spiders) were commonly observed near guttation drops. Sit-and-wait crab spiders may visit these drops to hunt for prey visiting the drops (figure 1d). Pollinators, such as honeybees (Apidae), were also observed visiting guttation drops in highbush blueberries.

## (c) Guttation effects on insect community composition

The multivariate PCA showed that the insect community differs in composition between those captured on sticky traps near plants with guttation versus those captured on traps near control plants (without guttation), with PC1 and PC2 explaining 46.4% of the total variance (figure 5a,b). Subsequent univariate analysis showed that captures of predators and parasitoids increased on traps near plants with guttation compared with those without guttation (figure 6). In particular, the predatory gall midges, long-legged flies and parasitic wasps increased by 33.3%, 18.4%, and 28.3%, respectively (figure 6b; electronic supplementary material, table S6).

In contrast with the findings on predators and parasitoids, guttation did not increase the attraction of insect herbivores (figure 6c). In fact, the number of aphids (Aphididae) was 35.6% lower on traps near plants with guttation (figure 6d; electronic supplementary material, table S5), possibly due to the increased abundance of their natural enemies in plants with guttation. Similarly, the numbers of mosquitoes (Culicidae) were 40.4% lower on traps near plants with guttation (figure 6g; electronic supplementary material, table S6).

## 4. Discussion

For the first time, this study documents the benefits to insects provided by guttation as a nutrient-rich plant-derived food source. Here, we demonstrate that plant guttation droplets are, unlike nectar, present in an ecosystem during an entire growing season, attract insects from numerous taxa, increase twice the abundance of predators and parasitoids in plants with droplets versus plants without droplets, enhance the survival and reproductive capacity of insects from three distinct families and feeding lifestyles (i.e. herbivores, parasitic wasps and predators) and are not only rich in carbohydrates but also contain proteins.

Until now, nectar, pollen, extrafloral nectar and honeydew have been the main supplementary plant-derived food sources described for insects in ecosystems [46,47]. However, the presence of these four food sources can vary considerably [46], making their availability unpredictable for a foraging insect. By contrast, in highbush blueberry fields, guttation was present throughout the day and during the entire growing season. More than 5% of the blueberry leaves had at least one guttation drop throughout the season. The greatest percentage of leaves with guttation was observed during the phenological stage of shoot growth and expansion and early fruit development (green fruit and beginning of fruit colouring). This finding agrees with previous reports

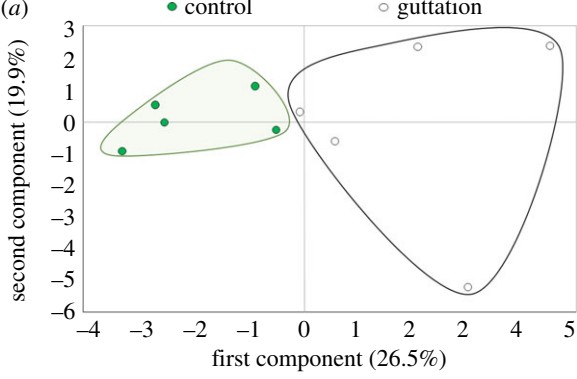

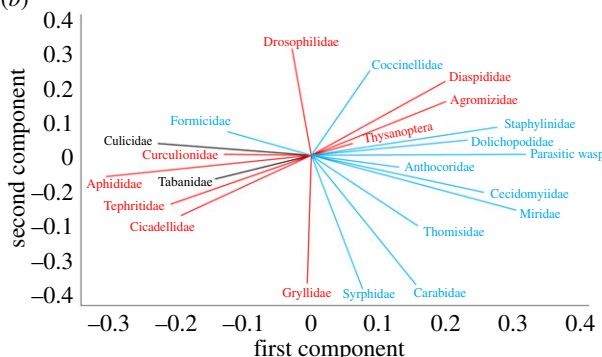

**Figure 5.** PCA for the effects of guttation on the communities of insects caught on sticky traps in a highbush blueberry field. (a) Score plot. Control, traps near plants without guttation; guttation, traps near plants with guttation. The first two axes of the PCA account for 46.4% of the total data variation. (b) Loading plot. In red are shown the herbivores (families), in blue are shown predators and parasitic wasps and in black are other insects.

indicating that guttation fluid in crops is present mostly during the development of young leaves [16,30,38,48–50]. Also, abiotic factors such as high RH [15], daily fluctuations in ambient temperatures [51] and high soil temperature [32] are known to regulate the occurrence of guttation fluids in other crops [15]. We found that guttation in highbush blueberries was positively correlated with ambient and soil temperatures but not with RH or solar radiation.

Guttation droplets were visited by numerous insect species with different feeding lifestyles. For instance, insect predators such as gall midges, long-legged flies, hoverflies, ants, lacewings and crab spiders were either observed near guttation drops or captured on traps near guttation drops. Our chemical analyses confirmed that guttation droplets contain a high concentration of sugars but also contain proteins. Until now, there has been only six studies (including ours) that analysed the nutritional (sugar and protein) content of guttation in plants. These studies find a wide range of sugar (from 0.0000271 to 1.5 g ml$^{-1}$) and protein (from 0.0027 to 30 mg ml$^{-1}$) concentrations in guttation droplets across four and 10 plant species, respectively (electronic supplementary material, table S7). According to these studies, sugar (1.5 g ml$^{-1}$) and protein (4.3 mg ml$^{-1}$) content in the guttation of highbush blueberries is higher than those of most other plant species. Thus, in highbush blueberries, the occurrence of guttation drops during the entire season and the content of sugars and proteins make them a highly available, nutrient-rich food source for insects in this ecosystem. The presence of sugars and proteins in guttation droplets

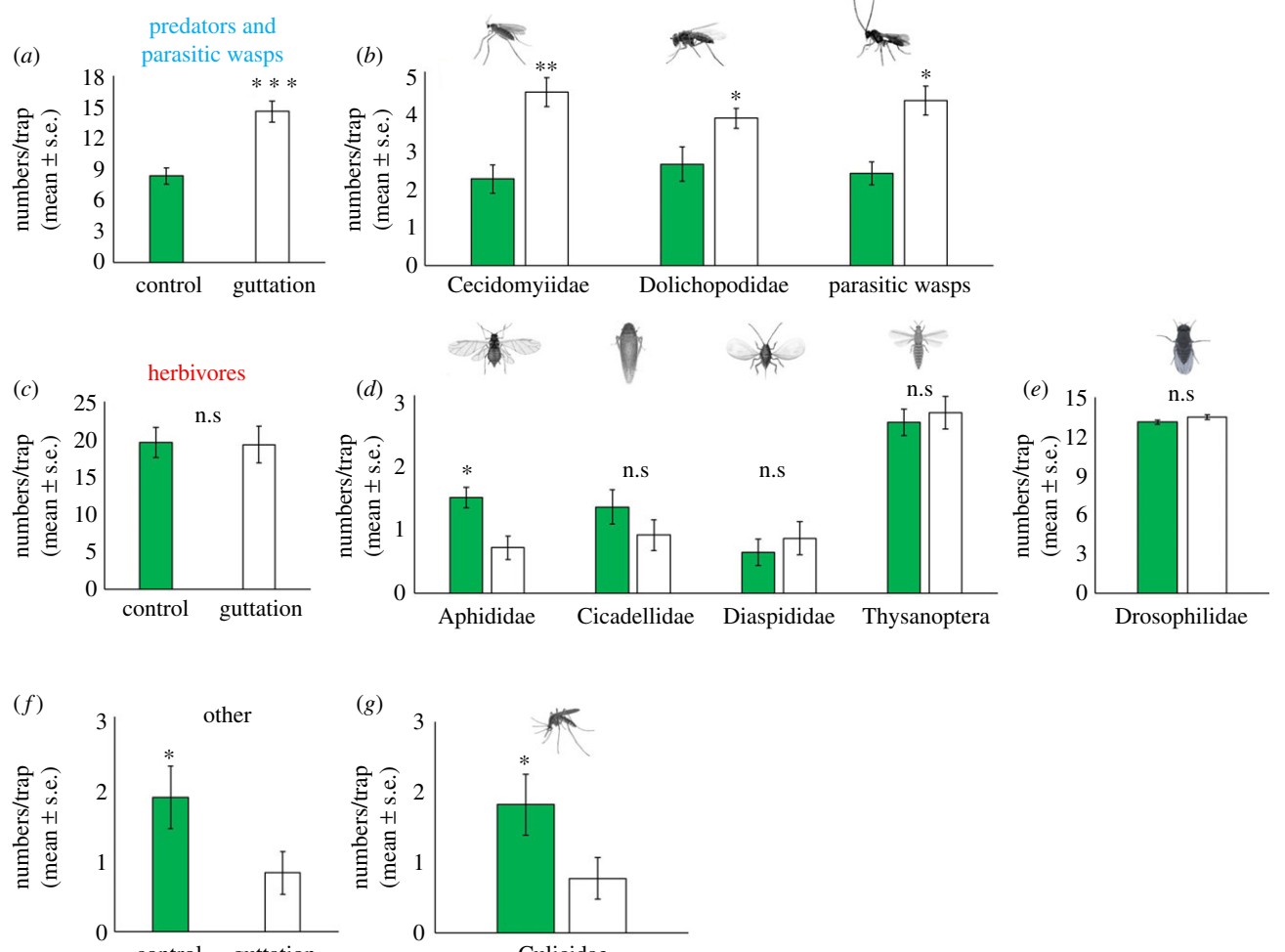

**Figure 6.** Predators and parasitic wasps (*a,b*), herbivores (*c–e*) and other insects (*f,g*) captured on sticky traps either near a highbush blueberry plant with no guttation (control) or a plant. *$p < 0.05$, **$p < 0.01$, ***$p < 0.001$ according to Student's *t*-tests; n.s., not significant. (Online version in colour.)

was previously documented in other plants, and the presence of proteins has been related to plant defence mechanisms against pathogens [17,24,39]. In fact, guttation fluids contain the same nutrients transported to sites of active vegetative and reproductive growth for the formation and development of vegetative tissue and fruits in plants [16,17,27] but in a more concentrated shape of a drop, which explains its peak occurrence at the time of active leaf and fruit growth in highbush blueberries.

Our manipulative experiment, designed to compare the composition of the insect community in plants with and without guttation, showed that the presence of guttation increases the attraction of insect predators, such as Cecidomyiidae and Dolichopodidae flies, to plants. Many insect predators and parasitoids rely primarily or exclusively on carbohydrates for energy to fuel their daily physical activities and for metabolic upkeep [46]. Because carbohydrates are rapidly metabolized, they are an ideal source of energy during flight [52], particularly for insects with high-frequency wing beats, such as Diptera and Hymenoptera [53] that oxidize carbohydrates in their wing muscles [10]. This dependence on carbohydrates might explain the attraction of dipteran predators and parasitic wasps towards plants with guttation drops. Thus, the availability and reliability of guttation in an ecosystem may increase the fitness and biological control potential of these natural enemies, as well as pollinator fitness, when nectar is scarce.

In contrast with predators and parasitoids, guttation did not increase the attraction of herbivores. In fact, the number of aphids was lower on traps near plants with guttation, possibly due to the increased abundance of natural enemies in plants with guttation. Interestingly, the numbers of mosquitoes were also approximately 40% lower on traps near plants with guttation, which could be due to the presence of Thomisidae spiders near the guttation drops, as these spiders are known to use mosquitoes as prey [54]. Finally, we would like to highlight that the manipulative assay was carried out at the end of the season, when guttation drops were almost completely absent in the field, to avoid the natural presence of guttation drops. Although the insect community might be different at the end of the season than during the guttation season, flying predators and parasitoids were present in both periods (electronic supplementary material, table S5). Therefore, we consider that both of these insect groups can increase their abundance if guttation drops are present throughout the season.

Until this study, the benefits of guttation to insects were unknown. To address this gap, we assessed the fitness of an insect herbivore (*D. suzukii*), a parasitic wasp (*A. ervi*) and a predatory insect (*C. rufilabris*) when they had access to guttation drops. The adult longevity and egg load of these three insect species were higher when they fed on guttation as a food source than the sugar plus protein, protein-only or water-only diets but was similar to those fed on a

sugar-only diet. These results, together with the chemical analyses, suggest that sugars and proteins in guttation droplets contributed to the increase in longevity and fecundity of these three insect species. In the field, insect predators and parasitoids may find other carbohydrate and protein sources, such as nectar, mature fruits, honeydew, pollen and prey. The former three are rich carbohydrate sources but, compared to guttation, are ephemeral in blueberry fields. As a comparison, the nectar (sucrose) content in highbush blueberry (*V. corymbosum*) flowers ranges from 0.33–0.46 g ml$^{-1}$ [55], which is about three times lower than the sugar content in the leaf guttation of this crop (1.5 g ml$^{-1}$) that we report in our study. Future studies are, however, needed to compare the nutritional benefits of these two food sources on the fitness of insect predators and parasitoids. To obtain proteins, adult insect predators and parasitoids can feed on their hosts and prey. Therefore, the contribution of guttation to their protein diet might not be as important as it is as a carbohydrate source, unless hosts and prey are scarce. This is especially important for parasitoid species that need to feed on proteins to mature eggs. They could maintain their egg load by feeding on guttation drops when food sources are limited. Although our field experiment showed that insect herbivores were not attracted to plants with guttation droplets, the dipteran pest *D. suzukii* also benefited when it fed on them. Therefore, as occurs with other plant-derived food sources [56], herbivorous pests could benefit from the guttation drops, resulting in a potential ecological cost for the plant.

Guttation can be a rich food source for insects, but it can become contaminated by systemic insecticides (e.g. neonicotinoids). The accumulation of insecticides in guttation drops is another negative consequence of using systemic insecticides because pollinators and insect predators that feed on it might die (e.g. [34,35,37,57]). This route of insecticide exposure should be considered by the environmental agencies, especially in crops and seasons where nectar is scarce because many insects may search for guttation drops on which to feed. In highbush blueberries, as well as in many other crops, neonicotinoids are recommended to control aphids (foliar applications) and soil grubs (soil application) and are applied in early June, which coincides with peak guttation occurrence. Whether these insecticides are present in guttation drops after application and affect insect predators and parasitoids will be the subject of future studies.

In conclusion, this study demonstrates that guttation is a nutrient-rich food source for insects that is available throughout the daytime and during the entire growing season in an ecosystem like highbush blueberry. Insects from a wide variety of taxa and feeding lifestyles visit and are attracted to guttation droplets. Moreover, both plant mutualistic and antagonistic insects using guttation as part of their diet may benefit by extending their lifespan and fecundity. Our study provides the first evidence that plant guttation constitutes an important, but yet underexplored, trait in plants with profound implications on its interactions with insects in ecosystems. It also highlights the need to consider guttation as an important plant-derived source of nutrients for insects in cropping systems as well as in non-crop communities and ecosystems.

Data accessibility. The datasets supporting this article have been uploaded as part of the electronic supplementary material.

Authors' contributions. P.U.-B. and C.R.-S. conceived the ideas; P.U.-B., C.R.-S. and A.T. designed the experiments; P.U.-B. collected the data; J.G.-C. analysed carbohydrate and protein contents; P.U.-B., C.R.-S. and A.T. analysed the data; P.U.-B. and C.R.-S. wrote the first draft of the manuscript. All authors revised and approved the final version of the manuscript.

Competing interests. The authors have no competing interests.

Funding. The study was partially supported by funding from the USDA NIFA Specialty Crops Research Initiative (SCRI) programme (award no. 2015-51181-24252), the USDA NIFA Organic Agriculture Research and Extension Initiative (OREI) programme (award no. 2018-51300-28434), the New Jersey Blueberry Research Council and the Hatch projects nos. NJ08252 and NJ08140 to C.R.-S. J.G.-C. was supported by the Ramón y Cajal Program (RYC-2013-13834).

Acknowledgements. We thank Vanessa Garzón, Stephanie Aponte and Giovana Jimenez for laboratory and field assistance. We also thank two anonymous reviewers for their constructive comments on an earlier draft of the manuscript.

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
