## [Reviewer comments · Proceedings of the Royal Society B: Biological Sciences]

Review History

RSPB-2020-1080.R0 (Original submission)

Review form: Reviewer 1

Recommendation

Accept with minor revision (please list in comments)

Scientific importance: Is the manuscript an original and important contribution to its field?

Excellent

General interest: Is the paper of sufficient general interest?

Good

Quality of the paper: Is the overall quality of the paper suitable?

Excellent

Is the length of the paper justified?

Yes

Should the paper be seen by a specialist statistical reviewer?

No

Do you have any concerns about statistical analyses in this paper? If so, please specify them explicitly in your report.

No

It is a condition of publication that authors make their supporting data, code and materials available - either as supplementary material or hosted in an external repository. Please rate, if applicable, the supporting data on the following criteria.

Is it accessible?

Yes

Is it clear?

Yes

Is it adequate?

Yes

Do you have any ethical concerns with this paper?

No

Comments to the Author

I must say, I was delighted to review this well prepared manuscript.

The authors present a solid study that investigates the nutritional effects of guttation drops in blueberry fields on different insect taxa. They combine field observations and manipulative field experiments with fecundity and survival assays in the lab to investigate the influence of guttation drops on insect performance and insect community composition.

The manuscript is very well prepared, the research questions are clear, the methods (including the statistical analyses) are sound and presented in sufficient detail but not overwhelming. The results section is short and precise, and the discussion does not overinterpret the findings of the study.

Apart from nectar research, publications that investigate plant secretions, exudates or plant sap as insect food sources are scarce. Therefore this manuscript is clearly of interest to the field. The use of different methodologies and the overarching approach to use chemical analyses, performance assays, and insect community observations makes it also appealing for a broad audience.

Thus, I think this manuscript is suitable for publication in Proceedings of the Royal Society B

I have some minor concerns I hope the authors will be able to address:

Line 157: "water only, sugar only (1 M solution), protein only (1 M yeast extract), sugar plus protein (1:1),"

why were these protein and sugar concentrations chosen for the diet? Do they fit to the amounts measured in guttation droplets?

Line 225 "To avoid competition and increase insect attraction, the experiment was conducted at the time when leaves had hardened and guttation drops were almost completely absent in the field"

I fully understand this reasoning, but I have a concern here. At this time, can you exclude that the attracted insect community changed due to seasonal variation? In other words, is the insect

community that you attracted with your manipulated guttation drops late in the season similar to the community that was there a month before when guttation was frequent in the field? If not, can you make an argument why your conclusions about the effects of guttation droplets in the field still apply to the community that is actually present when the guttation drops normally appear?

Line 263: "1.5 ± 0.2 g/ml"

You mean mg/ml, right?

Line 362: "Because insects were captured on traps before they had physical contact with the guttation drops, it is likely that they used visual and/or chemical cues to locate them or to assess danger"

From the explanation in the methods section, this part is not clear for me. If the transparent trap-card was hung into the plant, why can the insects not have been caught after leaving a guttation drop?

Line 371: "suggest that sugars and proteins in guttation droplets may have contributed to the increase in longevity and fecundity of these three insects and confirm that guttation is a rich source of nutrients for them."

I generally agree with the conclusion, but I would suggest to rephrase this part for clarity and maybe to increase the level of confidence in the first part of the sentence but to reduce it in the second part. Sugar does increase longevity, that is absolutely clear. But adding protein to the sugar seems to reduce the longevity in most taxa. I would speculate that the yeast extract used in the diet might not be optimal for these insects. I would suggest to address this shortly. Addition of yeast extract still led to some increase in egg load of the females, but considering the shortened lifespan under this diet, it is hard to guess the overall fecundity of the insects, had they lived longer when supplied with a not harmful protein mix. It is also unclear what the "usual" fecundity under field conditions or under optimal nutritional conditions is. Without at least an idea of what might be the potential/realistic fecundity of the insects, I think it is hard to make a statement about how important the guttation drops are for it. Might it not be the case that all diets offered in this experiment are suboptimal compared to all the food sources available under field conditions? Clearly these insects visit guttation droplets, but can you exclude that they feed on nectar or on other food sources as well? If not, how do you know that guttation drops are a rich source of nutrients for them in real life? Can you maybe compare the amounts of protein and sugar in guttation drops to the amounts of nectar or another more investigated important food source for insects to support your case?

Line 379: "A negative consequence of feeding on guttation drops in agroecosystems is the fact that systemic insecticides can be accumulated in these drops"

I think that the accumulation of insecticides in guttation drops is a negative consequence of using the insecticides, not a negative effect of insects feeding on guttation drops.

Review form: Reviewer 2

Recommendation

Major revision is needed (please make suggestions in comments)

Scientific importance: Is the manuscript an original and important contribution to its field?

Excellent

General interest: Is the paper of sufficient general interest?

Excellent

Quality of the paper: Is the overall quality of the paper suitable?

Good

Is the length of the paper justified?

Yes

Should the paper be seen by a specialist statistical reviewer?

No

Do you have any concerns about statistical analyses in this paper? If so, please specify them explicitly in your report.

No

It is a condition of publication that authors make their supporting data, code and materials available - either as supplementary material or hosted in an external repository. Please rate, if applicable, the supporting data on the following criteria.

Is it accessible?

N/A

Is it clear?

N/A

Is it adequate?

N/A

Do you have any ethical concerns with this paper?

No

Comments to the Author

I was excited to read this paper when I saw the title...and after reading it I'm consider it to be an outstanding and novel piece of work. However, I think it can be improved. Below I provide a number of suggestions that I believe will focus and tighten the results and interpretations.

A big comment is that the paper is for Proceedings...but in places it reads as though it is targeted for an applied ecology journal. I strongly encourage the authors to be mindful of this throughout the entire manuscript. In the title for example, remove agroecosystems. Why not just go with "Plant guttation provides nutrient-rich food for insects"? Additionally, throughout the manuscript, the authors refer to "beneficial" insects (parasitoids and lacewings). Drop this description throughout the manuscript. It distracts from the broader message.

Abstract...line 26...three different feeding guilds. I don't like this. Say three different feeding lifestyles...an herbivore, a parasitoid and a predator.

Line 30...beneficial insects...don't like this. Specifically say parasitoid and predator abundance increased.

Line 33...I'd drop pollination and biological control. Focus on the big picture of plant-insect interactions.

Line 74...as in the abstract...tell us the three lifestyles

Paragraph starting at line 99...the species names get confusing here. Maybe use the common names when describing the players (e.g. line 99 and 103).

Lines 189-215 could be improved for clarity. I had to re-read these lines multiple times.

Lines 262...these is great data...but how does it compare to other plant tissues (nectar, extrafloral nectaries, xylem, phloem).

Line 264 (and the rest of the ms as well). In all cases only adult insects are examined. This really needs to come out earlier. Emphasized in the introduction for a start.

Line 275...more than 5% of the blueberry leaves had at least one guttation drop during the season. This is very unsatisfying. So maybe a leaf only had one drop ever? This doesn't seem right. I'm guessing guttation drops were common. I think the authors need to do a better job of explaining the patterns of guttation, including how many leaves at one time had guttation drops. I feel like starting with the 5% number downplays what is really going on. I don't think it will take much to present this data in a slightly more compelling way.

Line 283...1,057 insects visited guttation drops. What is this relative to? Is this number high, low? There is no context.

Lines 285-287...provide common names whenever possible as many people reading this may not be familiar the family names. Also be consistent...for example using "parasitic wasps" in the same sentence with a bunch of family names.

Line 287...beneficial...do not use!

Line 298...beneficial

Line 309...how significant is this benefit in terms of a number or numbers?

Line 316...reorder...nectar, pollen, extrafloral nectar and honeydew

Line 320...this more than 5% number appears. How many guttation drops/plant occur each day? How abundant is this resource? Once a guttation drop is removed that day...does it come back?

Line 329-331...remove this sentence...not appropriate for this journal.

Line 333-336...clean this up...need better delivery of this information.

Line 338...what does it mean to be "reliable"?

Line 340...change crop to plant

Line 348...natural enemies (remove this type of description throughout)

Line 355...agroecosystem...just ecosystem

Line 368...how does this relate to the field?

Line 372...may have contributed...seems weak.

Line 374...synovigenic - avoid jargon

Lines 379-386...drop all of this. Not needed for this journal.

Figure 3...need to improve this figure. I suggest connecting the data points for each category.

Figure 4...Found it challenging to decipher this figure

Figure 6...rename beneficial to parasitoids

Decision letter (RSPB-2020-1080.R0)

19-Jun-2020

Dear Dr Urbaneja-Bernat:

Your manuscript has now been peer reviewed and the reviews have been assessed by an Associate Editor. The reviewers' comments (not including confidential comments to the Editor) and the comments from the Associate Editor are included at the end of this email for your reference. As you will see, the reviewers and the Editors have raised some concerns with your manuscript and we would like to invite you to revise your manuscript to address them.

Research ethics:

Use of animals and field studies:

Please submit a copy of your revised paper within three weeks. If we do not hear from you within this time your manuscript will be rejected. If you are unable to meet this deadline please let us know as soon as possible, as we may be able to grant a short extension.

Best wishes,
Dr Sasha Dall
mailto: proceedingsb@royalsociety.org

Associate Editor
Board Member: 1
Comments to Author:

We have received two reviews of RSPB-2020-1080, "Plant guttation provides nutrient-rich food for insects with implications for their communities in agroecosystem". Both reviewers are positive about the novelty, importance and rigour of the study, but they also have a number of critically constructive suggestions, which all seem reasonable. I recommend the authors revise the paper to carefully and thoroughly address all of the reviewers' comments. If the authors disagree with any of the suggestions, they should articulate why they have not made a change.

I also make two additional suggestions to the reviewers. First, Table 1 should be moved to the supplement; the results of this table can be concisely summarized in the main text. Second, and more substantial, I think the impact of the paper would be even greater if they analyze the metabolic content of guttation droplets from a diverse selection of additional plant species (e.g. 5-10 spp). Their results are novel, but it will be of even greater interest to Proc B's broad audience if you convince readers these results are not specific to just *Vaccinium corymbosum*, but to guttation droplets more generally, which are widespread among plant species. If the authors find that guttation droplets frequently contain carbohydrates and proteins, then it suggests this is a very widespread and important pattern that has been completely overlooked by previous research studying plant-insect interactions. I suggest that the diverse collection of plants include plants from the following clades of angiosperms: 1) Asterids (ideally 2 more since *Vaccinium* is an Asterid), 2) Rosids, 3) Vitales, 4) Caryophyllales, 5) Saxifrageles, 6) Ranunculid, 7) a monocot or two. To be clear, I am not suggesting the whole slew of additional experiments in every species, but a simple metabolic profile of guttation droplets from a few replicate plants from each species. I do not see this additional suggestion as completely necessary for meeting the criterion for being acceptable for publication, but it would take the paper to a higher impact level, so I leave it with the authors to decide whether this extra investment of time is feasible.

Sincerely,
 Marc Johnson
 Associate Editor, Proceedings B
 University of Toronto Mississauga

Reviewer(s)' Comments to Author:

Referee: 1

Comments to the Author(s)

I must say, I was delighted to review this well prepared manuscript.

The authors present a solid study that investigates the nutritional effects of guttation drops in blueberry fields on different insect taxa. They combine field observations and manipulative field experiments with fecundity and survival assays in the lab to investigate the influence of guttation drops on insect performance and insect community composition.

The manuscript is very well prepared, the research questions are clear, the methods (including the statistical analyses) are sound and presented in sufficient detail but not overwhelming. The results section is short and precise, and the discussion does not overinterpret the findings of the study.

Apart from nectar research, publications that investigate plant secretions, exudates or plant sap as insect food sources are scarce. Therefore this manuscript is clearly of interest to the field. The use of different methodologies and the overarching approach to use chemical analyses, performance assays, and insect community observations makes it also appealing for a broad audience.

Thus, I think this manuscript is suitable for publication in Proceedings of the Royal Society B

I have some minor concerns I hope the authors will be able to address:

Line 157: "water only, sugar only (1 M solution), protein only (1 M yeast extract), sugar plus protein (1:1),"

why were these protein and sugar concentrations chosen for the diet? Do they fit to the amounts measured in guttation droplets?

Line 225 “To avoid competition and increase insect attraction, the experiment was conducted at the time when leaves had hardened and guttation drops were almost completely absent in the field”

I fully understand this reasoning, but I have a concern here. At this time, can you exclude that the attracted insect community changed due to seasonal variation? In other words, is the insect community that you attracted with your manipulated guttation drops late in the season similar to the community that was there a month before when guttation was frequent in the field? If not, can you make an argument why your conclusions about the effects of guttation droplets in the field still apply to the community that is actually present when the guttation drops normally appear?

Line 263: “ 1.5 ± 0.2 g/ml”
You mean mg/ml, right?

Line 362: “Because insects were captured on traps before they had physical contact with the guttation drops, it is likely that they used visual and/or chemical cues to locate them or to assess danger”

From the explanation in the methods section, this part is not clear for me. If the transparent trapcard was hung into the plant, why can the insects not have been caught after leaving a guttation drop?

Line 371: “suggest that sugars and proteins in guttation droplets may have contributed to the increase in longevity and fecundity of these three insects and confirm that guttation is a rich source of nutrients for them.”

I generally agree with the conclusion, but I would suggest to rephrase this part for clarity and maybe to increase the level of confidence in the first part of the sentence but to reduce it in the second part. Sugar does increase longevity, that is absolutely clear. But adding protein to the sugar seems to reduce the longevity in most taxa. I would speculate that the yeast extract used in the diet might not be optimal for these insects. I would suggest to address this shortly. Addition of yeast extract still led to some increase in egg load of the females, but considering the shortened lifespan under this diet, it is hard to guess the overall fecundity of the insects, had they lived longer when supplied with a not harmful protein mix. It is also unclear what the “usual” fecundity under field conditions or under optimal nutritional conditions is. Without at least an idea of what might be the potential/realistic fecundity of the insects, I think it is hard to make a statement about how important the guttation drops are for it. Might it not be the case that all diets offered in this experiment are suboptimal compared to all the food sources available under field conditions? Clearly these insects visit guttation droplets, but can you exclude that they feed on nectar or on other food sources as well? If not, how do you know that guttation drops are a rich source of nutrients for them in real life? Can you maybe compare the amounts of protein and sugar in guttation drops to the amounts of nectar or another more investigated important food source for insects to support your case?

Line 379: “A negative consequence of feeding on guttation drops in agroecosystems is the fact that systemic insecticides can be accumulated in these drops”

I think that the accumulation of insecticides in guttation drops is a negative consequence of using the insecticides, not a negative effect of insects feeding on guttation drops.

Referee: 2

Comments to the Author(s)

I was excited to read this paper when I saw the title...and after reading it I consider it to be an outstanding and novel piece of work. However, I think it can be improved. Below I provide a number of suggestions that I believe will focus and tighten the results and interpretations.

A big comment is that the paper is for Proceedings...but in places it reads as though it is targeted for an applied ecology journal. I strongly encourage the authors to be mindful of this throughout the entire manuscript. In the title for example, remove agroecosystems. Why not just go with "Plant guttation provides nutrient-rich food for insects"? Additionally, throughout the manuscript, the authors refer to "beneficial" insects (parasitoids and lacewings). Drop this description throughout the manuscript. It distracts from the broader message.

Abstract...line 26...three different feeding guilds. I don't like this. Say three different feeding lifestyles...an herbivore, a parasitoid and a predator.

Line 30...beneficial insects...don't like this. Specifically say parasitoid and predator abundance increased.

Line 33...I'd drop pollination and biological control. Focus on the big picture of plant-insect interactions.

Line 74...as in the abstract...tell us the three lifestyles

Paragraph starting at line 99...the species names get confusing here. Maybe use the common names when describing the players (e.g. line 99 and 103).

Lines 189-215 could be improved for clarity. I had to re-read these lines multiple times.

Lines 262...these is great data...but how does it compare to other plant tissues (nectar, extrafloral nectaries, xylem, phloem).

Line 264 (and the rest of the ms as well). In all cases only adult insects are examined. This really needs to come out earlier. Emphasized in the introduction for a start.

Line 275...more than 5% of the blueberry leaves had at least one guttation drop during the season. This is very unsatisfying. So maybe a leaf only had one drop ever? This doesn't seem right. I'm guessing guttation drops were common. I think the authors need to do a better job of explaining the patterns of guttation, including how many leaves at one time had guttation drops. I feel like starting with the 5% number downplays what is really going on. I don't think it will take much to present this data in a slightly more compelling way.

Line 283...1,057 insects visited guttation drops. What is this relative to? Is this number high, low? There is no context.

Lines 285-287...provide common names whenever possible as many people reading this may not be familiar the family names. Also be consistent...for example using "parasitic wasps" in the same sentence with a bunch of family names.

Line 287...beneficial...do not use!

Line 298...beneficial

Line 309...how significant is this benefit in terms of a number or numbers?

Line 316...reorder...nectar, pollen, extrafloral nectar and honeydew

Line 320...this more than 5% number appears. How many guttation drops/plant occur each day? How abundant is this resource? Once a guttation drop is removed that day...does it come back?

Line 329-331...remove this sentence...not appropriate for this journal.

Line 333-336...clean this up..need better delivery of this information.

Line 338...what does it mean to be “reliable”?

Line 340...change crop to plant

Line 348...natural enemies (remove this type of description throughout)

Line 355...agroecosystem...just ecosystem

Line 368...how does this relate to the field?

Line 372...may have contributed...seems weak.

Line 374...synovigenic – avoid jargon

Lines 379-386...drop all of this. Not needed for this journal.

Figure 3...need to improve this figure. I suggest connecting the data points for each category.

Figure 4...Found it challenging to decipher this figure

Figure 6...rename beneficial to parasitoids

Author's Response to Decision Letter for (RSPB-2020-1080.R0)

See Appendix A.

Decision letter (RSPB-2020-1080.R1)

27-Jul-2020

Dear Dr Urbaneja-Bernat:

Your manuscript has now been peer reviewed and the reviews have been assessed by an Associate Editor. The reviewers' comments (not including confidential comments to the Editor) and the comments from the Associate Editor are included at the end of this email for your reference. As you will see, the reviewers and the Editors have raised some concerns with your manuscript and we would like to invite you to revise your manuscript to address them.

To submit your revision please log into <http://mc.manuscriptcentral.com/prsb> and enter your Author Centre, where you will find your manuscript title listed under "Manuscripts with

Decisions." Under "Actions", click on "Create a Revision". Your manuscript number has been appended to denote a revision.

Research ethics:

Use of animals and field studies:

It is a condition of publication that you make available the data and research materials supporting the results in the article (<https://royalsociety.org/journals/authors/author-guidelines/#data>). Datasets should be deposited in an appropriate publicly available repository and details of the associated accession number, link or DOI to the datasets must be included in the Data Accessibility section of the article (<https://royalsociety.org/journals/ethics-policies/data-sharing-mining/>). Reference(s) to datasets should also be included in the reference list of the article with DOIs (where available).

All supplementary materials accompanying an accepted article will be treated as in their final form. They will be published alongside the paper on the journal website and posted on the online figshare repository. Files on figshare will be made available approximately one week before the

accompanying article so that the supplementary material can be attributed a unique DOI. Please try to submit all supplementary material as a single file.

Please submit a copy of your revised paper within three weeks. If we do not hear from you within this time your manuscript will be rejected. If you are unable to meet this deadline please let us know as soon as possible, as we may be able to grant a short extension.

Best wishes,
Dr Sasha Dall
Editor, Proceedings B
mailto:proceedingsb@royalsociety.org

Associate Editor
Comments to Author:

I have now considered the revised version of RSPB-2020-1080.R1, "Plant guttation provides nutrient-rich food for insects", for consideration in Proceedings B. I maintain that this is a high quality study that reports novel results that could be acceptable for Proceedings B. While the authors have made a number of revisions to their paper in line with the reviewers' comments, I regret that they still have not addressed several important points that are necessary for the paper to be considered for publication. I recommend the authors be given one further opportunity to address these outstanding issues, which I describe below.

1) The reviewers still have not addressed one of my major points in the first version of this paper, which was as follows: "Their results are novel, but it will be of even greater interest to Proc B's broad audience if you convince readers these results are not specific to just *Vaccinium corymbosum*".

It is unfortunate the authors did not take the opportunity analyze the sugar and protein content of guttation droplets from additional plant species. This result would have elevated the novelty and generality of their results beyond the single system they studied. This suggestion would not have required them to have conduct additional experiments; it could have been as easy as collecting droplets from wild plants and determining the range of values that sugar and protein content among species, including where *V. corymbosum* fell within that distribution. I recognize access to labs might be restricted at the moment, but I had hoped the authors would accomplish this as the sugar and protein assays seem relatively straightforward as described in the methods. But as I mentioned before, I did not see this inclusion as a necessary criterion for publication. Nevertheless, the issue still remains that the importance of this work is cast too narrowly. Readers have no sense from the information provided on whether these results are specific to *V. corymbosum*, or whether they could be more general across all plants that exude droplets through guttation, including both crop and wild species. At minimum, the authors should include a paragraph in the Discussion discussing what is known about the metabolic contents of guttation droplets, especially as it relates to sugar and protein content, and whether they expect similar effects on arthropod populations and communities from other systems. For example, Singh 2016 Bot Rev and Singh and Singh 2013 Phytochem Review both provide numerous references where such data can be obtained. The authors should then discuss whether *V. corymbosum* falls within this range. What would be most convincing is a supplemental 2-D

figure, with sugar concentration plotted on the x-axis, and protein concentration plotted on the y-axis, and a point with x and y- error bars included for each species for which protein and sugar concentrations have been quantified.

2) My comment is also related to the following comment from reviewer 2 that was not sufficiently addressed:

“ A big comment is that the paper is for Proceedings...but in places it reads as though it is targeted for an applied ecology journal. I strongly encourage the authors to be mindful of this throughout the entire manuscript. In the title for example, remove agroecosystems. Why not just go with “Plant guttation provides nutrient-rich food for insects”? Additionally, throughout the manuscript, the authors refer to “beneficial” insects (parasitoids and lacewings). Drop this description throughout the manuscript. It distracts from the broader message.”

The manuscript still reads as if it is targeted to an agricultural journal, and while Proceedings B does publish papers with applied relevance, Proc B's broad readership mostly consists of scientists working on fundamental problems (i.e., non-applied) in biology. While the authors did change the title, the revisions to the paper were insufficient to broaden the scope of the importance of the paper to appeal to Proc B's readership. A naïve reader may come away with thinking that these results are only relevant to *V. corymbosum*, or only to crops. But guttation droplets are much more widespread among plants and this paper would be considerably more appealing to Proc B's broad readership if the authors could convince them that this is likely to be a general phenomenon among plants that exude guttation droplets.

Examples of where I think the authors have cast the importance of their work too narrowly include:

Line 21 (abstract): “most of the world's major crops”
How about plants in general!?

Line 57: “guttation droplets ... have been reported in most major crops ...”
What about non-crop plants?

Line 350 (also see line 353): “This finding agrees with previous reports indicating that guttation fluid in crops ...”
What about non-crop systems?

Line 441: “It also highlights the need to consider guttation as an important plant-derived source of nutrients for insects in cropping systems.”
What about in non-crop communities and ecosystems?

It is appropriate for the authors to discuss how these results are relevant to crop systems, but it is equally important to convince readers that these results are important in non-crop systems. The good news is that this is an easy fix.

3) The authors have not fully addressed one of reviewer 1's comments:
“Do they fit to the amounts measured in guttation droplets?”

Starting on line 145, please also express the concentrations of sucrose and protein in each assay diet in the same units they are expressed on line 291 of the results, so that the experimental manipulations and the measurements from plants are easily compared by readers.

4) A final issue is that a number of typos and grammatical errors have crept into the latest version of the paper. It is important that the authors carefully read their paper and remove these errors. I have attached a draft of the paper where I use Adobe's editing feature to highlight and/or annotate a number of these issues.

If the authors are able to carefully and fully address these remaining comments I would be happy to recommend their paper for publication in Proceedings B.

Sincerely,
Marc Johnson
Associate Editor,
Proceedings B

Author's Response to Decision Letter for (RSPB-2020-1080.R1)

See Appendix B.

Decision letter (RSPB-2020-1080.R2)

21-Aug-2020

Dear Dr Urbaneja-Bernat

I am pleased to inform you that your manuscript RSPB-2020-1080.R2 entitled "Plant guttation provides nutrient-rich food for insects" has been accepted for publication in Proceedings B.

The referee(s) have recommended publication, but also suggest some minor revisions to your manuscript. Therefore, I invite you to respond to the referee(s)' comments and revise your manuscript. Because the schedule for publication is very tight, it is a condition of publication that you submit the revised version of your manuscript within 7 days. If you do not think you will be able to meet this date please let us know.

1) A text file of the manuscript (doc, txt, rtf or tex), including the references, tables (including captions) and figure captions. Please remove any tracked changes from the text before submission. PDF files are not an accepted format for the "Main Document".

2) A separate electronic file of each figure (tiff, EPS or print-quality PDF preferred). The format should be produced directly from original creation package, or original software format. PowerPoint files are not accepted.

3) Electronic supplementary material: this should be contained in a separate file and where possible, all ESM should be combined into a single file. All supplementary materials accompanying an accepted article will be treated as in their final form. They will be published alongside the paper on the journal website and posted on the online figshare repository. Files on figshare will be made available approximately one week before the accompanying article so that the supplementary material can be attributed a unique DOI.

Sincerely,
Dr Sasha Dall
Editor, Proceedings B
<mailto:proceedingsb@royalsociety.org>

Associate Editor:
Board Member
Comments to Author:

The authors have done a very good job at addressing the most recent revisions. I am prepared to recommend the paper for acceptance, but the authors misunderstood the following comment:

2) My comment is also related to the following comment from reviewer 2 that was not sufficiently addressed:

REVIEWER COMMENT: " A big comment is that the paper is for Proceedings ... Additionally, throughout the manuscript, the authors refer to "beneficial" insects (parasitoids and lacewings). Drop this description throughout the manuscript. It distracts from the broader message."

AUTHOR RESPONSE: We fixed this throughout the manuscript. We have changed the title to make it more general. We have also avoided referring to "predators and parasitic wasps" and instead use "insects" and "beneficial insects" to make it more general, wherever it applies.

The change to the title was as suggested, but the request was to NOT use the term "beneficial insects". "Predators and parasitoids" is clearer and more appropriate, so they should change the revision back to the version in R1. I am sorry for any confusion on this point.

With this last very minor change, I am happy to recommend the paper be accepted to Proceedings B and I wish the authors congratulations on the achievement that this paper represents.

Sincerely,
Marc Johnson
Associate Editor, Proceedings B
University of Toronto Mississauga

Author's Response to Decision Letter for (RSPB-2020-1080.R2)

See Appendix C.

Decision letter (RSPB-2020-1080.R3)

24-Aug-2020

Dear Dr Urbaneja-Bernat

I am pleased to inform you that your manuscript entitled "Plant guttation provides nutrient-rich food for insects" has been accepted for publication in Proceedings B.

Open Access

Paper charges

Sincerely,

Appendix A

July 16, 2020

Dear Dr. Marc Johnson
Associate Editor, Proceedings of the Royal Society B

On behalf of my co-authors, I am writing to provide a revised version of our manuscript (RSPB-2020-1080) entitled "Plant guttation provides nutrient-rich food for insects." The reviews provided were very helpful for improving the text, and we have addressed the concerns below in a point-by-point response, with our responses provided in italics. Black letters = reviewer; Red letters = author.

Associate Editor

I also make two additional suggestions to the reviewers. First, Table 1 should be moved to the supplement; the results of this table can be concisely summarized in the main text. Second, and more substantial, I think the impact of the paper would be even greater if they analyze the metabolic content of guttation droplets from a diverse selection of additional plant species (e.g. 5-10spp). Their results are novel, but it will be of even greater interest to Proc B's broad audience if you convince readers these results are not specific to just *Vaccinium corymbosum*, but to guttation droplets more generally, which are widespread among plant species. If the authors find that guttation droplets frequently contain carbohydrates and proteins, then it suggests this is a very widespread and important pattern that has been completely overlooked by previous research studying plant-insect interactions. I suggest that the diverse collection of plants include plants from the following clades of angiosperms: 1) Asterids (ideally 2 more since *Vaccinium* is an Asterid), 2) Rosids, 3) Vitales, 4) Caryophyllales, 5) Saxifrageles, 6) Ranunculid, 7) a monocot or two. To be clear, I am not suggesting the whole sweet of additional experiments in every species, but a simple metabolic profile of guttation droplets from a few replicate plants from each species. I do not see this additional suggestion as completely necessary for meeting the criterion for being acceptable for publication, but it would take the paper to a higher impact level, so I leave it with the authors to decide whether this extra investment of time is feasible.

We appreciate your kindly and constructive suggestions.

We agree to move "Table 1" to the supplementary material. It is now "Table 5S". The main results of this table are concisely summarized in the main text.

The suggestion by the Associate Editor to add more plant species that produce plant guttation in this study is a great one. However, because of time constraints and current situation with a global pandemic, which is limiting our research capacities, we cannot add more species to this study. Adding more plant species to the study is not trivial since it will require growing plants under the same environmental conditions for likely several months and then take the samples for analyses and analyze them which will add additional months. We strongly feel, and the reviewers agree based on their comments, that the study as presented is strong enough to demonstrate that guttation can be a food source for insects. We are taking seriously the suggestion by the Associate Editor because we consider this to be an important subject and plan to collect guttation from other plant species and study their effects on insects but this will be the subject of future publications.

Reviewer 1

I must say, I was delighted to review this well prepared manuscript. The authors present a solid study that investigates the nutritional effects of guttation drops in blueberry fields on different

insect taxa. They combine field observations and manipulative field experiments with fecundity and survival assays in the lab to investigate the influence of guttation drops on insect performance and insect community composition.

We thank the reviewer for the positive comments on our paper.

The manuscript is very well prepared, the research questions are clear, the methods (including the statistical analyses) are sound and presented in sufficient detail but not overwhelming. The results section is short and precise, and the discussion does not overinterpret the findings of the study.

We thank the reviewer for the positive comments on our paper.

Apart from nectar research, publications that investigate plant secretions, exudates or plant sap as insect food sources are scarce. Therefore this manuscript is clearly of interest to the field. The use of different methodologies and the overarching approach to use chemical analyses, performance assays, and insect community observations makes it also appealing for a broad audience. Thus, I think this manuscript is suitable for publication in Proceedings of the Royal Society B.

We were delighted to read that Reviewer 1 found the study interesting and solid. We also thank Reviewer 1 for the time dedicated to review the manuscript and the constructive comments.

I have some minor concerns I hope the authors will be able to address:

Line 157: "water only, sugar only (1 M solution), protein only (1 M yeast extract), sugar plus protein (1:1)," why were these protein and sugar concentrations chosen for the diet?

Our aim was to compare guttation drops with diets rich on sugars and proteins. We selected the concentrations based on previous studies. We have added this clarification in the manuscript: "Sugar and protein diets were used as control diets to confirm their positive effects on the longevity and fecundity of the three insects. The concentrations used in the assay were selected based on previous studies (Benelli et al. 2017; Urbaneja-Bernat et al. 2013)." (lines 134-136)

Do they fit to the amounts measured in guttation droplets?

We did not know the sugar and protein concentrations in the guttation droplets when we started these bioassays. We suspected that it was a rich sugar source because it was sticky, but we did not know it would contain proteins! We realized that guttation drops had proteins when we analyzed the egg maturation experiment. To avoid confusion and follow the chronological order of the assays, we have moved the sentence with the concentrations at the end of the paragraph (lines 265-266). We have also moved it in the M&M section (lines 159-178).

Line 225 "To avoid competition and increase insect attraction, the experiment was conducted at the time when leaves had hardened, and guttation drops were almost completely absent in the field" I fully understand this reasoning, but I have a concern here. At this time, can you exclude that the attracted insect community changed due to seasonal variation? In other words, is the insect community that you attracted with your manipulated guttation drops late in the season similar to the community that was there a month before when guttation was frequent in the field? If not, can you make an argument why your conclusions about the effects

of guttation droplets in the field still apply to the community that is actually present when the guttation drops normally appear?

We agree that insect communities might have changed between the first (Reliability of guttation as food source for insects in the field) and second experiment (Guttation effects on insect community composition). However, we cannot confirm this hypothesis because we used different sampling techniques: direct observations in the first experiment and sticky traps in the second. Despite this, parasitic wasps and flying predators were more abundant in plants with guttation in the second assay and they were also present in the first experiment. Therefore, we consider that the main conclusion still applies to the insect community present in the first experiment, when guttation was more common.

To address this, we have included in the Discussion section: “Finally, we would like to highlight that the manipulative assay was carried out at the end of the season, when guttation drops were almost completely absent in the field, to avoid the natural presence of guttation drops. Although the insect community might be different at the end of the season than during guttation season, parasitic wasps and flying predators were present in both periods (Table S5). Therefore, we consider that both beneficial insect groups can increase their abundance if guttation drops are present throughout the season.” (lines 356-362)

Line 263: “ 1.5 ± 0.2 g/ml”. You mean mg/ml, right?

This value, although seems high, is the right one. We show below the process that we followed to calculate sugar concentration:

1)

Samples	m(total_sugar) (μ g/vial)	[total_sugar] mg/ml
1	21.06	105.32
2	9.91	49.53
3	13.73	68.65
4	9.90	49.50
5	10.77	53.84
6	18.85	94.23
7	16.12	80.58
8	8.71	43.53
9	9.35	46.76
10	12.71	63.56

* We have divided by 10 to have the concentration per μ l in the diluted sample (I used 10 μ l to make the determination), then we multiplied by the dilution made (50) and we have the concentration in mg / ml of the received solution.

2)

Rep	Number of drops	Total vol drops (μ L)	Total vol drops (μ L) + PBS (150 μ L)
1	86.00	5.83	155.83
2	92.00	5.82	155.82
3	144.00	5.83	155.83

4	123.00	5.72	155.72
5	121.00	7.69	157.69
6	97.00	7.52	157.52
7	207.00	8.22	158.22
8	100.00	7.10	157.10
9	133.00	7.41	157.41
10	161.00	7.72	157.72

Samples	[total_sugar] mg/ml
1	2816.70
2	1325.93
3	1835.09
4	1347.74
5	1103.83
6	1973.99
7	1550.30
8	963.21
9	993.67
10	1298.46

* In this case we multiplied the concentration in the received solution by the total volume and obtained the mass of sugar, then we divided that mass by the original volume of guttation and obtained the concentration of sugars in the drops excreted by the plant. We have made all these data available in the "Data accessibility" xls document.

Line362: "Because insects were captured on traps before they had physical contact with the guttation drops, it is likely that they used visual and/or chemical cues to locate them or to assess danger" From the explanation in the methods section, this part is not clear for me. If the transparent trap-card was hung into the plant, why can the insects not have been caught after leaving a guttation drop?

We agree, insects could have been caught after feeding. We have removed this speculative sentence (lines 419-423)

Line 371: "suggest that sugars and proteins in guttation droplets may have contributed to the increase in longevity and fecundity of these three insects and confirm that guttation is a rich source of nutrients for them." I generally agree with the conclusion, but I would suggest to rephrase this part for clarity and maybe to increase the level of confidence in the first part of the sentence but to reduce it in the second part. Sugar does increase longevity, that is absolutely clear. But adding protein to the sugar seems to reduce the longevity in most taxa. I would speculate that the yeast extract used in the diet might not be optimal for these insects. I would suggest to address this shortly. Addition of yeast extract still led to some increase in egg load of the females, but considering the shortened lifespan under this diet, it is hard to guess the overall fecundity of the insects, had they lived longer when supplied with a not harmful protein mix. It is also unclear what the "usual" fecundity under field conditions or under

optimal nutritional conditions is. Without at least an idea of what might be the potential/realistic fecundity of the insects, I think it is hard to make a statement about how important the guttation drops are for it. Might it not be the case that all diets offered in this experiment are suboptimal compared to all the food sources available under field conditions? Clearly these insects visit guttation droplets, but can you exclude that they feed on nectar or on other food sources as well? If not, how do you know that guttation drops are a rich source of nutrients for them in real life? Can you maybe compare the amounts of protein and sugar in guttation drops to the amounts of nectar or another more investigated important food source for insects to support your case?

We have removed the second part of the sentence and have included the implications of our results in real life (this has been also suggested by Reviewer 2), where insects can find other food sources. "These results, together with the chemical analyses, suggest that sugars and proteins in guttation droplets contributed to the increase in longevity and fecundity of these three insect species. In the field, parasitic wasps and predators may find other carbohydrate and protein sources, such as nectar, mature fruits, honeydew, pollen, and prey. The former three are rich carbohydrate sources but, compared to guttation, are ephemeral in blueberry fields. To obtain proteins, adult parasitic wasps and predators can feed on their hosts and prey. Therefore, the contribution of guttation to their protein diet might not be as important as it is as a carbohydrate source, unless hosts and prey are scarce." (lines: 368-379)

Line 379: "A negative consequence of feeding on guttation drops in agroecosystems is the fact that systemic insecticides can be accumulated in these drops". I think that the accumulation of insecticides in guttation drops is a negative consequence of using the insecticides, not a negative effect of insects feeding on guttation drops.

Thanks for the comment. We agree with the reviewer and have modified the sentence accordingly to: "Guttation can be a rich food source for insects but it can become contaminated by systemic insecticides (e.g. neonicotinoids). The accumulation of insecticides in guttation drops is another negative consequence of using systemic insecticides because predators, parasitic wasps and pollinators that feed on it might die (e.g. [29,30,32,49]). This route of insecticide exposure should be considered by the environmental agencies, especially in crops and seasons where nectar is scarce because many insects may search for guttation drops on which to feed" (lines 386-391)

Reviewer 2

I was excited to read this paper when I saw the title...and after reading it I'm consider it to be an outstanding and novel piece of work. However, I think it can be improved. Below I provide a number of suggestions that I believe will focus and tighten the results and interpretations.

We are delighted to read that Reviewer 2 found our study outstanding and novel. We also thank Reviewer 2 for the time dedicated to review the manuscript and the constructive comments.

A big comment is that the paper is for Proceedings...but in places it reads as though it is targeted for an applied ecology journal. I strongly encourage the authors to be mindful of this throughout the entire manuscript. In the title for example, remove agroecosystems. Why not just go with "Plant guttation provides nutrient-rich food for insects"? Additionally, throughout

the manuscript, the authors refer to “beneficial” insects (parasitoids and lacewings). Drop this description throughout the manuscript. It distracts from the broader message.

As suggested by the reviewer, we have changed the title to: “Plant guttation provides nutrient-rich food for insects”. We have also edited the manuscript following the reviewer’s recommendation.

Abstract...line 26...three different feeding guilds. I don’t like this. Say three different feeding lifestyles...an herbivore, a parasitoid and a predator.

We have changed as suggested. (line 25)

Line 30...beneficial insects...don’t like this. Specifically say parasitoid and predator abundance increased.

We have changed this sentence as suggested. (line 30)

Line 33...I’d drop pollination and biological control. Focus on the big picture of plant-insect interactions.

We have removed “pollination and biological control”. (line 34)

Line 74...as in the abstract...tell us the three lifestyles

We have added the three lifestyles (line 80-81)

Paragraph starting at line 99...the species names get confusing here. Maybe use the common names when describing the players (e.g. line 99 and 103).

*Because the common names for *Aphidius ervi* and *Chrysoperla rufilabris* are too generic (i.e., parasitic wasps and green lacewings, respectively), we prefer to keep the species names.*

Lines 189-215 could be improved for clarity. I had to re-read these lines multiple times.

We have added some additional information to clarify the methods used. (line 223-230)

Lines 262...these is great data...but how does it compare to other plant tissues (nectar, extrafloral nectaries, xylem, phloem).

*Thanks for the comment. We have added some additional information: “ As a comparison, the nectar (sucrose) content in highbush blueberry (*V. corymbosum*) flowers ranges from 0.33–0.46 g/ml [50], which is about three times lower than the sugar content in the leaf guttation of this crop (1.5 g/ml) that we report in our study. Future studies are, however, needed to compare the nutritional benefits of these two food sources on the fitness of parasitic wasps and predators.” (line 372-377)*

Line 264 (and the rest of the ms as well). In all cases only adult insects are examined. This really needs to come out earlier. Emphasized in the introduction for a start.

We added this information in M&M section (Line 87: “adults of”)

Line 275...more than 5% of the blueberry leaves had at least one guttation drop during the season. This is very unsatisfying. So maybe a leaf only had one drop ever? This doesn’t seem right. I’m guessing guttation drops were common. I think the authors need to do a better job

of explaining the patterns of guttation, including how many leaves at one time had guttation drops. I feel like starting with the 5% number downplays what is really going on. I don't think it will take much to present this data in a slightly more compelling way.

Thanks for this recommendation. We had been very conservative explaining our results. Following the recommendation, we have changed to: "In highbush blueberry fields, plant guttation was present throughout the day and during the entire growing season (Figure 4; Table S3). The highest percentage of leaves with guttation was observed during the phenological stage of shoot growth and expansion and early fruit development (green fruit and beginning of fruit coloring), when at noon more than 40% of the leaves had guttation drops (Figure 4a)." (lines 269-273)

Line 283...1,057 insects visited guttation drops. What is this relative to? Is this number high, low? There is no context.

We removed the total number of insects visiting guttation drops, and we added "a broad variety of arthropod fauna" (lines 278)

Lines 285-287...provide common names whenever possible as many people reading this may not be familiar the family names. Also be consistent...for example using "parasitic wasps" in the same sentence with a bunch of family names.

We provide common names to make easiest to the reader through the text.

Line 287...beneficial...do not use!

Changed.

Line 298...beneficial

Changed.

Line 309...how significant is this benefit in terms of a number or numbers?

In the following lines, we have added that the presence of guttation drops increased twice the number of beneficial insects. We have not detailed the increase of longevity and fecundity because they depended on the sex and species.

Line 316...reorder...nectar, pollen, extrafloral nectar and honeydew

Changed.

Line 320...this more than 5% number appears. How many guttation drops/plant occur each day? How abundant is this resource? Once a guttation drop is removed that day...does it come back?

To answer the first question, we would need to count the total number of leaves per each sampled highbush blueberry each time that we sampled the guttation presence and drops. Although, we only sampled a few number of leaves per bush (100), we think that it is sufficient to understand the frequency of daily and seasonal occurrence of guttation. However, we find it difficult to reliably estimate the amount of guttation at a plant or field level since the number of young leaves per bush varies depending on age.

The answer for the second question is “yes”. For the laboratory experiments, we removed the drops from the leaves and approximately 2 days after, we observed new drops. However, we did not collect data on this so more research is needed to understand the rate of production of guttation droplets. This will be the subject of future studies.

Line 329-331...remove this sentence...not appropriate for this journal.

The sentence has been removed.

Line 333-336...clean this up..need better delivery of this information.

We have changed to: “Guttation droplets were visited by numerous insect species with different feeding lifestyles” (lines 327-328)

Line 338...what does it mean to be “reliable”?

We changed “reliable” to “highly available” (lines 333)

Line 340...change crop to plant

Changed.

Line 348...natural enemies (remove this type of description throughout)

Changed.

Line 355...agroecosystem...just ecosystem

Changed.

Line 368...how does this relate to the field?

Following the recommendations of both Reviewers, we relate these results with other food sources that they can find in the field (lines 370-378)

Line 372...may have contributed...seems weak.

We removed “may have” (line 369)

Line 374...synovigenic – avoid jargon

We have changed to: “This is especially important for parasitoid species that need to feed on proteins to mature eggs. They could maintain their egg load by feeding on guttation drops when hosts are limited.” (lines 379-381)

Lines 379-386...drop all of this. Not needed for this journal.

The negative effect of neonicotinoids for insects has been widely documented in Proceedings of the Royal Society b: Biological Sciences during the last 5 years. There are more than 20 papers published on this topic, including two reviews. Thus, we think that it is important to highlight that guttation can be contaminated by systemic insecticides such as neonicotinoids because many insects may search and feed on guttation when nectar is scarce. Please, see that we have modified this paragraph following the recommendations of Reviewer 1.

Figure 3...need to improve this figure. I suggest connecting the data points for each category.

We cannot follow this recommendation because insects were killed and dissected to measure the egg load. Therefore, there is not continuity between points.

Figure 4...Found it challenging to decipher this figure

We agree this figure contains a lot of information: seasonal and daily presence of guttation drops and plant phenology. We prefer to keep them together but if the reviewers or editor have another suggestion, we will consider it.

Figure 6...rename beneficial to parasitoids

We have changed to parasitic wasps and predators.

Thank you

Sincerely,

Pablo Urbaneja-Bernat on behalf the rest of coauthors
Department of Entomology
Rutgers University
P.E. Marucci Center
email: paurbaneja@gmail.com

Appendix B

Associate Editor

Comments to Author:

1) The reviewers still have not addressed one of my major points in the first version of this paper, which was as follows: “Their results are novel, but it will be of even greater interest to Proc B's broad audience if you convince readers these results are not specific to just *Vaccinium corymbosum*”.

It is unfortunate the authors did not take the opportunity analyze the sugar and protein content of guttation droplets from additional plant species. This result would have elevated the novelty and generality of their results beyond the single system they studied. This suggestion would not have required them to have conduct additional experiments; it could have been as easy as collecting droplets from wild plants and determining the range of values that sugar and protein content among species, including where *V. corymbosum* fell within that distribution. I recognize access to labs might be restricted at the moment, but I had hoped the authors would accomplish this as the sugar and protein assays seem relatively straightforward as described in the methods. But as I mentioned before, I did not see this inclusion as a necessary criterion for publication. Nevertheless, the issue still remains that the importance of this work is cast too narrowly. Readers have no sense from the information provided on whether these results are specific to *V. corymbosum*, or whether they could be more general across all plants that exude droplets through guttation, including both crop and wild species. At minimum, the authors should include a paragraph in the Discussion discussing what is known about the metabolic contents of guttation droplets, especially as it relates to sugar and protein content, and whether they expect similar effects on arthropod populations and communities from other systems. For example, Singh 2016 Bot Rev and Singh and Singh 2013 Phytochem Review both provide numerous references where such data can be obtained. The authors should then discuss whether *V. corymbosum* falls within this range. What would be most convincing is a supplemental 2-D figure, with sugar concentration plotted on the x-axis, and protein concentration plotted on the y-axis, and a point with x and y- error bars included for each species for which protein and sugar concentrations have been quantified.

Thank you for your comment. We have addressed your concern by following your second suggestion. We searched all published papers documenting the sugar and protein content of guttation. These data are included in Table S7. We have included in the Discussion a sentence that discusses these data in the context of our results. We decided to present the data in a table instead of a figure because the concentrations varied widely among plant species and this would make the figure less informative than a table.

2) My comment is also related to the following comment from reviewer 2 that was not sufficiently addressed:

“ A big comment is that the paper is for Proceedings...but in places it reads as though it is targeted for an applied ecology journal. I strongly encourage the authors to be mindful of this throughout the entire manuscript. In the title for example, remove agroecosystems. Why not just go with “Plant guttation provides nutrient-rich food for insects”? Additionally, throughout the manuscript, the authors refer to “beneficial” insects (parasitoids and lacewings). Drop this

description throughout the manuscript. It distracts from the broader message.”

We fixed this throughout the manuscript. We have changed the title to make it more general. We have also avoided referring to “predators and parasitic wasps” and instead use “insects” and “beneficial insects” to make it more general, wherever it applies.

The manuscript still reads as if it is targeted to an agricultural journal, and while Proceedings B does publish papers with applied relevance, Proc B’s broad readership mostly consists of scientists working on fundamental problems (i.e., non-applied) in biology. While the authors did change the title, the revisions to the paper were insufficient to broaden the scope of the importance of the paper to appeal to Proc B’s readership. A naïve reader may come away with thinking that these results are only relevant to *V. corymbosum*, or only to crops. But guttation droplets are much more widespread among plants and this paper would be considerably more appealing to Proc B’s broad readership if the authors could convince them that this is likely to be a general phenomenon among plants that exude guttation droplets.

Examples of where I think the authors have cast the importance of their work too narrowly include:

Line 21 (abstract): “most of the world’s major crops”
How about plants in general!?

To date, studies on guttation have been done in cropping systems. To address the editor’s comment we have deleted “most of the world’s major crops” and made the sentence more general.

Line 57: “guttation droplets ... have been reported in most major crops ...”
What about non-crop plants?

We have modified the sentence to avoid the focus on crops (lines 58-60).

Line 350 (also see line 353): “This finding agrees with previous reports indicating that guttation fluid in crops ...”
What about non-crop systems?

As indicated previously, most of our knowledge on guttation comes from cropping systems. Despite this, we feel that guttation is relevant for plants in general. To address the editor’s comments, we have toned down the focus on crops but unfortunately all the data on guttation content comes from crops, so we report these data in the Discussion and Table S7.

Line 441: “It also highlights the need to consider guttation as an important plant-derived source of nutrients for insects in cropping systems.”
What about in non-crop communities and ecosystems?

Thank you for the suggestion. We have changed this sentence to read “It also highlights the need to consider guttation as an important plant-derived source of nutrients for insects in cropping systems as well as in non-crop communities and ecosystems.”

3) The authors have not fully addressed one of reviewer 1's comments:

"Do they fit to the amounts measured in guttation droplets?"

Starting on line 145, please also express the concentrations of sucrose and protein in each assay diet in the same units they are expressed on line 291 of the results, so that the experimental manipulations and the measurements from plants are easily compared by readers.

We have indicated the guttation droplets contain sugar and protein. The concentration is reported in the Results section, so we refer the reader to this section for details.

4) A final issue is that a number of typos and grammatical errors have crept into the latest version of the paper. It is important that the authors carefully read their paper and remove these errors. I have attached a draft of the paper where I use Adobe's editing feature to highlight and/or annotate a number of these issues.

We have carefully read our paper and removed these errors. Also, we incorporated all edits made by the Associate Editor in the pdf document.

If the authors are able to carefully and fully address these remaining comments I would be happy to recommend their paper for publication in Proceedings B.

We have carefully addressed all the Associate Editor's comments in the revised manuscript and hope that the editor finds our revised paper suitable now for publication in Proceedings B.

Appendix C

Associate Editor:

Board Member

Comments to Author:

The authors have done a very good job at addressing the most recent revisions. I am prepared to recommend the paper for acceptance, but the authors misunderstood the following comment:

2) My comment is also related to the following comment from reviewer 2 that was not sufficiently addressed:

REVIEWER COMMENT: " A big comment is that the paper is for Proceedings ... Additionally, throughout the manuscript, the authors refer to "beneficial" insects (parasitoids and lacewings). Drop this description throughout the manuscript. It distracts from the broader message."

AUTHOR RESPONSE: We fixed this throughout the manuscript. We have changed the title to make it more general. We have also avoided referring to "predators and parasitic wasps" and instead use "insects" and "beneficial insects" to make it more general, wherever it applies.

The change to the title was as suggested, but the request was to NOT use the term "beneficial insects". "Predators and parasitoids" is clearer and more appropriate, so they should change the revision back to the version in R1. I am sorry for any confusion on this point.

With this last very minor change, I am happy to recommend the paper be accepted to Proceedings B and I wish the authors congratulations on the achievement that this paper represents.

Dear editor,

We are very grateful for the manuscript acceptance, and we appreciate and added your comments to improve our paper.

Consequently, we fixed the use of the term "beneficial insects" we have changed for "Predators and parasitoids" or "insect predators" throughout the manuscript (below the highlighted version of the manuscript).